# Patch-Wise and Keyword-Aware: Efficient Multi-Condition Control of Diffusion Transformers via Position-Aligned and Keyword-Scoped Attention

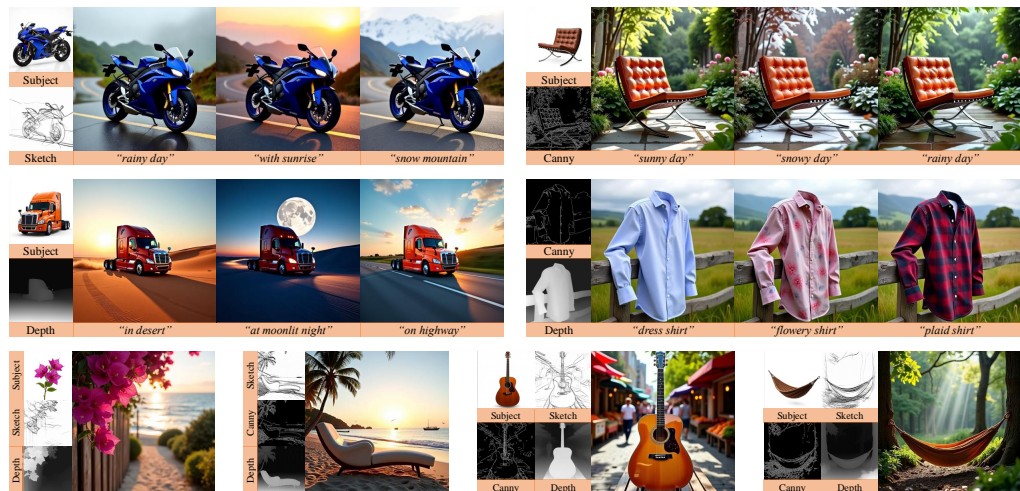

Figure 1: Visual results of our proposed PKA on multi-conditional generation. Our proposed PKA achieves high-quality multi-conditional generation with remarkable efficiency. Zoom in for better visualization.

## Abstract

While modern text-to-image models excel at generation from prompts, they often lack the fine-grained control necessary for specific user requirements like spatial layouts or subject appearances. Multi-condition control emerges as a key solution to this limitation. However, its application in Diffusion Transformers (DiTs) is severely hampered by the "concatenate-and-attend" strategy, which creates a prohibitive computational and memory bottleneck. Our analysis reveals that this computation is largely redundant. We therefore introduce Patch-wise and Keyword-Aware Attention (PKA), a framework using two specialized modules to eliminate this inefficiency. Position-Aligned Attention (PAA) confines spatial control to aligned patches, while Keyword-Scoped Attention (KSA) restricts subject-driven control to keyword-activated regions. Complemented by an early-timestep sampling strategy that accelerates training, PKA achieves up to a $10\times$ inference speedup and a $5.12\times$ reduction in attention module VRAM, all while maintaining or improving generative quality. Our work offers a practical path towards complex, fine-grained, and resource-friendly AI generation.

# 1 INTRODUCTION

After years of rapid development, Diffusion Transformers (DiTs) (Peebles & Xie, 2023; Esser et al., 2024) have become a leading architecture for image generation. While their performance is remarkable, most existing DiTs are guided predominantly by textual prompts. In many real-world scenarios, users often require more fine-grained control, such as specifying spatial arrangements, layouts, or visual references. This calls for multi-condition diffusion models that can flexibly incorporate both textural conditions and visual conditions.

In UNet-based diffusion models (Rombach et al., 2022; Podell et al., 2024), this challenge is typically addressed via feature-level fusion, as exemplified by methods like ControlNet (Zhang et al., 2023), in which different condition modalities are injected at various layers of the UNet via feature addition or modulation Sun et al. (2024); He et al. (2024; 2025). However, since feature fusion is less straightforward in transformer architectures, DiTs typically adopt a different paradigm: an attention-based interaction where all condition and noisy image tokens are concatenated and processed jointly (Tan et al., 2024; 2025; Wang et al., 2025; Pan et al., 2025).

However, this "concatenate-and-attend" strategy is computationally prohibitive. Assuming $c$ condition inputs and $n$ tokens per condition, the resulting attention computation scales as $O(c^2 n^2)$ due to the pairwise attention across all conditions and noisy image tokens at each transformer block. As the number of conditions increases (e.g., combining text, layout, reference image, and depth maps), the total sequence length grows substantially. The attention mechanism's computational and memory demands scale quadratically, creating a critical bottleneck that leads to excessive memory consumption and inference latency. This naturally forces a central question: *Does effective multi-condition control truly require such massive attention computation?*

To answer this question, we first investigated the attention patterns within existing multi-condition DiTs (Tan et al., 2024). Our analysis confirms that a significant portion of the attention computation is indeed redundant. This redundancy manifests differently depending on the condition type, which we categorize as spatial-aligned and subject-driven. For spatial-aligned conditions like layout maps, attention is intensely localized. As shown in Figure 2, the attention matrix is concentrated almost exclusively along its diagonal, indicating that only spatially aligned or adjacent patches interact meaningfully. Interactions between distant regions, in contrast, contribute negligible attention scores. For subject-driven conditions, such as textual descriptions of an object, attention is also sparse; only a small subset of cross attention map is strongly activated, and these activations correlate directly with the keyword-relevant areas of the image (Figure 3). This suggests that full attention is superfluous.

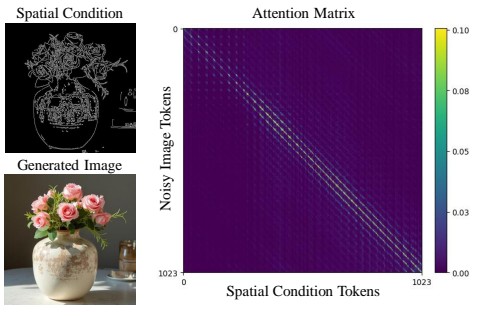
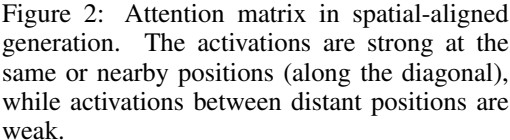
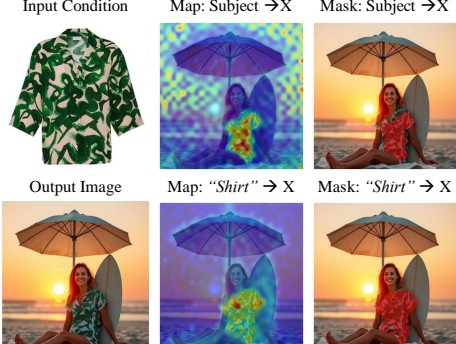

Figure 2: Attention matrix in spatial-aligned generation. The activations are strong at the same or nearby positions (along the diagonal), while activations between distant positions are weak.

Figure 3: Attention maps in subject-driven generation. Prompt: "On the beach, a lady wearing this shirt sits under a beach umbrella." X is the noisy image.

Motivated by these observations, we propose Patch-Wise and Keyword-Aware Attention (PKA), a novel mechanism for efficient multi-condition control. PKA leverages the inherent sparsity of these

attention patterns through two specialized, condition-aware modules designed to eliminate computational waste. The first, Position-Aligned Attention (PAA), addresses spatial-aligned conditions. It replaces full attention with a direct *one-to-one* correspondence between noisy image and condition tokens at the same spatial coordinates. By computing attention only between these aligned pairs, PAA enables highly localized control with minimal overhead. The second, Keyword-Scoped Attention (KSA), is designed for subject-driven conditions. It operates by first identifying the most relevant image regions via an attention map between the textual keyword and the noisy image tokens. This map is then used to create a relevance-scoped mask, confining subsequent attention computations only to these *salient regions* and drastically pruning the number of query-key interactions.

Furthermore, we posit that the conventional timestep sampling strategy employed in the training of flow matching (Lipman et al., 2023) models is suboptimal for fine-tuning multi-conditional generation tasks. To investigate the temporal influence of visual conditions, we conducted a perturbation analysis across the denoising process. This experiment revealed a crucial insight: visual conditions exert their strongest influence during the early, high-noise stages of generation. Motivated by this finding, we introduce a novel early-timestep sampling scheme that concentrates training on these critical phases, which accelerates convergence and enhances the final model's control fidelity.

By integrating these advancements, our experiments validate that we can significantly reduce both computational latency and the memory footprint of the attention mechanism, all without compromising the model's generative performance. Quantitatively, for scenarios with a high number of conditions, our method achieves an impressive speedup of up to $10.0\times$ and a $5.12\times$ reduction in memory consumption for the attention module.

In summary, our contributions are as follows.

- We conduct an in-depth analysis of multi-condition DiTs, identifying and characterizing the computational redundancy inherent in the standard full-attention mechanism.
- We propose methodological advancements to improve both inference and training efficiency, which include PKA, a lightweight attention framework to reduce computation, and an early timestep sampling strategy to accelerate fine-tuning convergence.
- We conduct comprehensive experiments, demonstrating that our method achieves state-of-the-art efficiency, including up to a $10\times$ speedup while maintaining or even improving generation quality and controllability over strong baselines.

## 2    RELATED WORK

### 2.1    CONTROLLABLE DIFFUSION GENERATION

Multi-condition generation enables users to guide the synthesis process with diverse inputs like spatial layouts or reference subjects. In UNet-based architectures, this is often achieved via feature-level fusion. One line of work, including ControlNet (Zhang et al., 2023), T2I-Adapter (Mou et al., 2024), and GLIGEN (Li et al., 2023), integrates spatial conditions like edge maps or poses through feature injection. Another line, featuring IP-Adapter (Ye et al., 2023), EZIGen (Duan et al., 2024), and InstantID (Wang et al., 2024), focuses on incorporating subject appearance from reference images to ensure identity consistency. In contrast, DiT-based models typically achieve multi-condition control through attention-based interaction. Frameworks like OminiControl (Tan et al., 2024) and UniCombine (Wang et al., 2025) have demonstrated the viability of this paradigm, where all conditional and latent tokens are concatenated for joint processing through full self-attention. However, this "concatenate-and-attend" approach faces a critical limitation: the computational cost grows quadratically with the number of tokens. This leads to substantial memory and runtime overhead, rendering these methods inefficient for practical scenarios that demand rich and varied conditional inputs.

### 2.2    EFFICIENT MECHANISM FOR DIFFUSION TRANSFORMERS

Several strategies have been proposed to mitigate computational overhead in DiTs. One research direction focuses on inference-time optimization, such as caching or decomposing less informative tokens (Zou et al., 2025; Ma et al., 2024; Zou et al., 2024; Liu et al., 2025; Chen et al., 2025).

Another popular approach improves efficiency by removing or simplifying layers that contribute minimally to the final generation quality (Fang et al., 2025; Zhu et al., 2024; Yang et al., 2025). For the specific task of multi-condition generation, methods like PixelPonder (Pan et al., 2025) and OminiControl2 (Tan et al., 2025) have also improved efficiency through techniques such as dynamic token pruning and input downsampling. In stark contrast, our PKA module reduces complexity from a different perspective: rather than relying on token reuse or architectural pruning, we leverage condition-specific structural priors to eliminate redundancy.

## 3 METHOD

### 3.1 PRELIMINARY

Diffusion Transformers (DiTs), such as FLUX.1 (Labs, 2024) and Stable Diffusion 3 (Esser et al., 2024), utilize a Transformer architecture as their denoising backbone. These models progressively refine noisy image tokens ($X \in \mathbb{R}^{N \times d}$), guided by various condition tokens like text ($C_T \in \mathbb{R}^{M \times d}$).

In multi-condition frameworks (Tan et al., 2024; 2025; Wang et al., 2025), additional visual condition tokens ($C_I \in \mathbb{R}^{N_I \times d}$) are incorporated by concatenating them with the text and image tokens. All tokens are then processed jointly through a multi-modal attention (MMA) mechanism:

$$\text{MMA}([C_T; X; C_I]) = \text{Softmax}\left(\frac{QK^\top}{\sqrt{d}}\right) V \tag{1}$$

The primary issue with this "concatenate-and-attend" paradigm is its computational cost. The attention matrix $QK^\top \in \mathbb{R}^{(M+N+N_I) \times (M+N+N_I)}$ scales quadratically with the sequence length, becoming prohibitively expensive as more conditions are added.

During training, these models typically use flow matching (Lipman et al., 2023) to learn the denoising process. Conventionally, the timestep $t$ for each training sample is drawn from a standard logit-normal distribution Logit-$\mathcal{N}(0, 1)$, ensuring the model is trained across all stages of the generation trajectory.

### 3.2 PATCH-WISE AND KEYWORD-AWARE ATTENTION

Building on the DiT-based text-to-image generation model FLUX (Labs, 2024), we propose Patch-Wise and Keyword-Aware Attention (PKA), a mechanism that decomposes the standard full-attention into a series of lightweight, specialized attentions. Our method operates on a sequence of tokens comprising text (T), the noisy image (X), the spatial condition (SP), and the subject condition (SJ). As illustrated in Figure 4(b), we fundamentally redesign the attention structure to reduce computational overhead. A key design principle is that condition tokens (SP and SJ) only perform self-attention within their respective conditions. This structural choice enables a highly efficient Condition Cache mechanism, as shown in Figure 4(a). The Key and Value projections for all condition tokens are computed only once in the first denoising step and are then cached and reused for all subsequent steps. This eliminates redundant computations across the denoising trajectory. The noisy image tokens (X) selectively interact with the conditions via our proposed Position-Aligned Attention (PAA) and Keyword-Scoped Attention (KSA) modules, while maintaining full attention with text (T).

#### 3.2.1 POSITION-ALIGNED ATTENTION

For the spatial condition, we introduce Position-Aligned Attention (PAA). The core intuition is that spatial layout primarily governs the structural arrangement of the image, and interactions between spatially distant patches are negligible. Therefore, it is both intuitive and efficient to compute attention only between corresponding spatial positions.

$$PAA\left([X; SP]\right)[i] = \text{Softmax}\left(\frac{Q_{X_i} K_{SP_i}^\top}{\sqrt{d}}\right) V_{SP_i} \tag{2}$$

As illustrated in Figure 4(c), we perform a one-to-one attention computation between the noisy image tokens and the spatial condition tokens at the same spatial coordinates. Specifically, we align

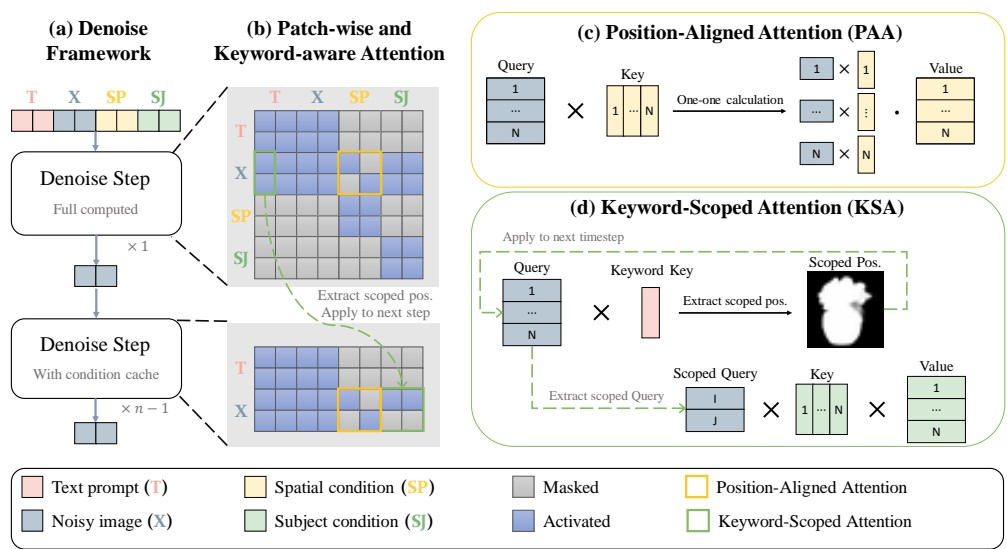

Figure 4: Overview of our method. (a) The denoise framework. Full computation occurs only at the first step; the Keys and Values of all condition tokens are then cached for subsequent steps. (b) Patch-Wise and Keyword-Aware Attention.Our decomposed attention mechanism, where conditions only perform self-attention (enabling the KV cache). The noisy image tokens (X) then interact with spatial (SP) and subject (SJ) conditions via PAA and KSA, respectively. (c) Position-Aligned Attention (PAA). PAA performs efficient one-to-one attention between the image (X) and spatial condition (SP) tokens at their aligned positions. (d) Keyword-Scoped Attention (KSA). KSA computes a relevance mask from text keywords in one step. This mask is then applied in subsequent steps to confine the attention computation between the image (X) and subject (SJ) to only the most relevant regions.

the $Q$, $K$, and $V$ representations at each position and compute their attention independently, as formulated in Eq. 2. This design reduces the computational complexity from $\mathcal{O}(N^2)$ in the full attention case to $\mathcal{O}(N)$, where $N$ is the number of tokens of the noisy image.

### 3.2.2 KEYWORD-SCOPED ATTENTION

For the subject condition, we propose Keyword-Scoped Attention (KSA). The key insight is that a subject's visual appearance is typically confined to a localized area within the generated image. Therefore, a global attention pass that computes interactions between the subject and all image tokens is inefficient and redundant.

To address this, KSA leverages temporal consistency (Zhou et al., 2025) in a two-step process, as illustrated in Figure 4(d). The first step, performed at timestep $t$, is to generate a binary mask $M_t$ that efficiently locates the subject. This is achieved by computing a lightweight attention map between the image queries $Q_X^t$ and the keys from a small set of keyword tokens $\mathbb{K}$:

$$M^t = \text{Norm}\left(\sum_{i\in\mathbb{K}}\left(Q_X^t K_i^{t\top}\right)\right) \geq \epsilon \tag{3}$$

Here, the keyword set $\mathbb{K}$ typically contains just 1 to 2 tokens, and $\epsilon$ is the mask threshold. Unless otherwise specified, we use $\epsilon = 0.2$ in the experiments.

According to the temporal consistency of the denoising process, we then reuse this mask $M$ at timestep $t + 1$ to select a subset of image tokens $\hat{Q}_X^{t+1} = Q_X^{t+1} \circ M^t$. By filtering out irrelevant positions beforehand, the final KSA attention computation is confined only to the semantically

meaningful regions, drastically reducing computational overhead:

$$KSA\left([X; SJ]\right) = \text{Softmax}\left(\frac{\hat{Q}_X^{t+1} K_{SJ}^{t+1}{}^\top}{\sqrt{d}}\right) V_{SJ}^{t+1} \tag{4}$$

### 3.3 EARLY-TIMESTEP SAMPLING

Prevailing flow matching models typically adopt a timestep sampling strategy where $t$ is drawn from a logit-normal distribution that $t \sim \text{Logit-}\mathcal{N}(0,1)$. However, our investigation reveals that this conventional approach is suboptimal for fine-tuning on multi-conditional control tasks. Our key empirical insight, as illustrated in Figure 5, is that the conditioning information is predominantly injected and learned during the initial phase of the denoising trajectory, i.e, higher $t$. To align the training process with this phenomenon, we propose a modified sampling strategy that intentionally prioritizes these critical early timesteps. We achieve this by skewing the sampling distribution towards the beginning of the process, drawing timesteps from a shifted logit-normal distribution: $t \sim \text{Logit-}\mathcal{N}(\mu, \delta)$, where $\mu > 0, \delta > 1$. This targeted approach concentrates the model's training on the temporal segments most crucial for effective conditional control.

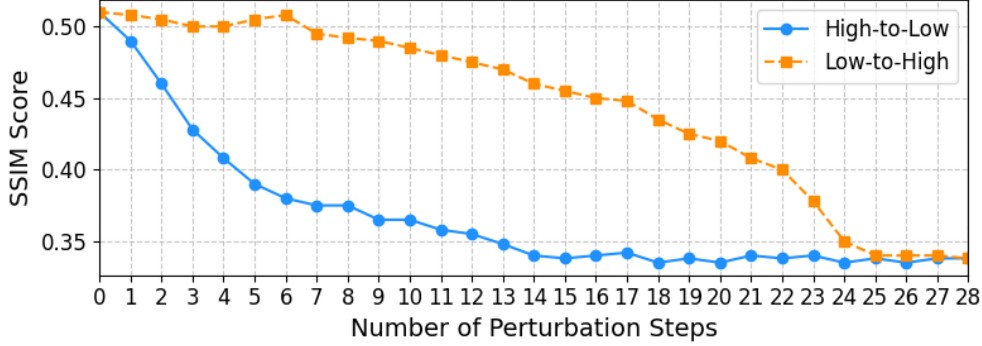

Figure 5: SSIM of Visual condition perturbation. "High-to-low" refers to applying perturbations sequentially from the early (high t) to late (low t) stages of the generation process, while "Low-to-high" is the reverse.

## 4 EXPERIMENT

### 4.1 SETUP

**Training Details.** We curate a subset from the Subject200K dataset (Tan et al., 2024), ensuring each image caption contains a descriptive keyword. This subset is then partitioned into training and testing sets. To ensure a fair comparison, we fine-tune the FLUX.1 (Labs, 2024) model using LoRA (Hu et al., 2022), which is trained for 20,000 iterations using the Prodigy (Mishchenko & Defazio, 2024) optimizer with a batch size of 1 and a gradient accumulation step of 4.

**Evaluation Details.** We employ OminiControl2 (Tan et al., 2025) and UniCombine (Wang et al., 2025) as baselines for our comparative analysis. Efficiency metrics, including inference latency and condition overhead, are measured on a single NVIDIA RTX 6000 Ada GPU. For evaluating generation quality, we define three multi-conditional tasks: Subject-Canny-to-Image, Subject-Depth-to-Image, and Canny-Depth-to-Image.

**Metrics.** To evaluate subject consistency, we calculate the CLIP-I (Radford et al., 2021) and DI-NOv2 (Oquab et al., 2024) scores between generated images and ground-truth images. To measure controllability, we compute the F1 Score for edge conditions and the MSE score for depth conditions

between maps extracted from the generated images and the original conditional inputs. For assessing generative quality, we compute FID (Heusel et al., 2017) and SSIM (Wang et al., 2004) between the generated and ground-truth image sets. Additionally, we adopt the CLIP-T (Radford et al., 2021) score to estimate the text consistency between the generated images and the text prompts.

## 4.2 MAIN RESULTS

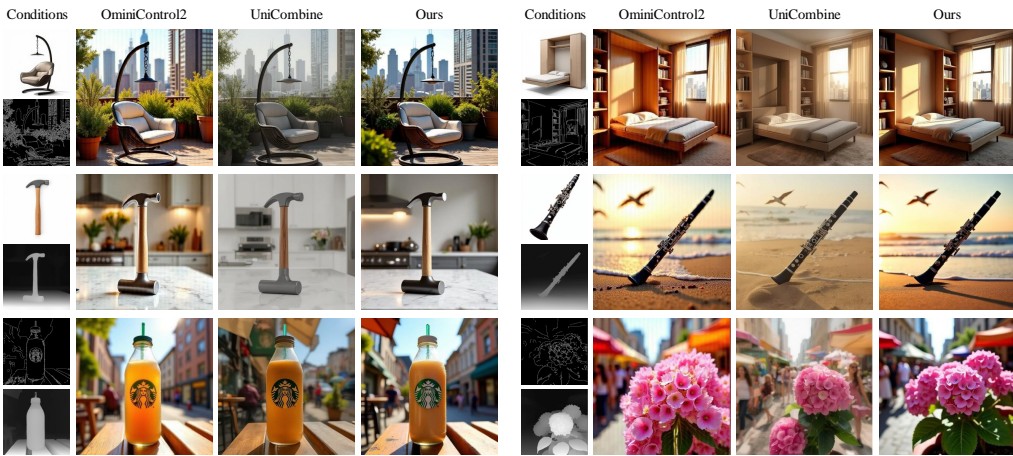

Figure 6: Qualitative comparison for multi-conditional control. From top to bottom: Subject-Canny-to-Image, Subject-Depth-to-Image, and Canny-Depth-to-Image. Zoom in for better visualization.

### 4.2.1 EFFICIENCY

Figure 7 illustrates the trend of inference time as the number of conditions increases. The results reveal that our method achieves a significant speedup, ranging from $3.90\times$ to $10\times$, compared to the full-attention mechanism in UniCombine. Notably, our approach also surpasses the performance of OminiControl2. In terms of memory efficiency, Figure 8 shows that our attention mechanism reduces the VRAM consumption by a factor of $2.46\times$ to $5.12\times$ relative to full attention. For this analysis, each condition is represented by 1024 tokens. Collectively, these findings demonstrate that our method substantially reduces computational costs in terms of both time and memory.

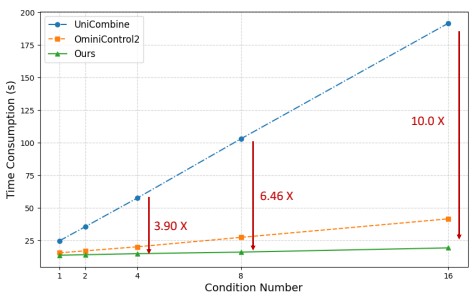

Figure 7: Time consumption comparison across different condition numbers.

Figure 8: VRAM consumption of attention mechanism comparison across different condition numbers.

### 4.2.2 QUALITATIVE COMPARISON

We evaluated our method on a suite of three challenging multi-conditional generation tasks: Subject-Canny-to-Image, Subject-Depth-to-Image, and Canny-Depth-to-Image. Figure 6 provides a quali-

tative comparison, showcasing the clear advantages of our approach over the baseline methods, OminiControl2 and UniCombine. While the performance gains are nuanced, our method consistently yields superior results. In direct comparison, images generated by OminiControl2 suffer from lower visual fidelity and noticeable artifacts. Meanwhile, UniCombine's outputs, though structurally coherent, often exhibit a muted or desaturated color palette, lacking the chromatic richness produced by our method.

### 4.2.3 QUANTITATIVE EVALUATION

The quantitative results in Table 1 confirm the effectiveness of our approach. Our method significantly outperforms competing baselines in Generative Quality and Subject Consistency across all tasks. In terms of Controllability, it is highly competitive, achieving the best results on most tasks, with the minor exception of a narrow margin on the Subject-Canny task. Furthermore, our model's Text Fidelity is comparable to the leading baseline, trailing by a perceptually negligible difference.

Table 1: Comparison of different methods across various tasks and metrics. The bold represents the optimal result.

| Task | Method | Quality | | Controllability | | Consistency | | Fidelity |
| | | FID↓ | SSIM↑ | F1↑ | MSE↓ | CLIP-I↑ | DINOv2↑ | CLIP-T↑ |
|---|---|---|---|---|---|---|---|---|
| Subject Canny | OminiControl2 | 72.03 | 0.406 | 0.192 | - | 0.878 | 0.867 | 0.327 |
| | UniCombine | 61.03 | 0.493 | **0.551** | - | 0.912 | 0.901 | **0.352** |
| | Ours | **52.99** | **0.553** | 0.414 | - | **0.945** | **0.926** | 0.349 |
| Subject Depth | OminiControl2 | 80.20 | 0.391 | - | 366 | 0.867 | 0.838 | 0.325 |
| | UniCombine | 70.22 | 0.454 | - | 312 | 0.911 | 0.879 | **0.350** |
| | Ours | **62.08** | **0.515** | - | **160** | **0.935** | **0.904** | 0.348 |
| Canny Depth | OminiControl2 | 71.87 | 0.475 | 0.194 | 303 | - | - | 0.342 |
| | UniCombine | 67.40 | 0.508 | 0.369 | 250 | - | - | **0.354** |
| | Ours | **53.01** | **0.613** | **0.411** | **114** | - | - | 0.353 |

## 4.3 ABLATION STUDY

### 4.3.1 EFFECT OF POSITION-ALIGNED ATTENTION

To evaluate our Position-Aligned Attention (PAA), we compare it against two baselines: full attention (W/o PAA), and Sliding Window Attention (SWA) (Pan et al., 2023) with various window sizes. While both methods produce high-fidelity images that adhere to the spatial conditions, as shown in Figure 9, our PAA architecture is demonstrably more efficient. PAA operates at a latency of just 13.63s and consumes only 237MB of VRAM, outperforming even the most efficient SWA (14.00s and 276MB). This confirms PAA delivers high-quality spatial control at substantially lower computational cost.

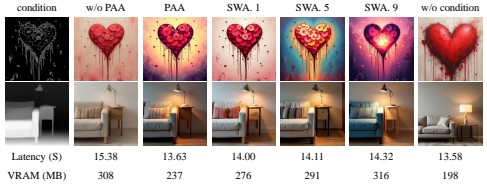
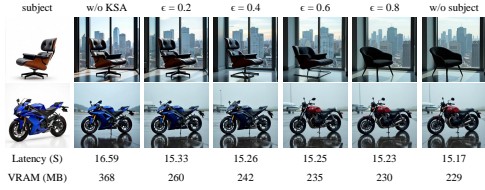

Figure 9: Ablation study on the PAA module. Zoom in for better visualization.

Figure 10: Ablation study on the KSA mask threshold $\epsilon$. Zoom in for better visualization.

### 4.3.2 EFFECT OF KEYWORD-SCOPED ATTENTION

Our Keyword-Scoped Attention (KSA) module provides powerful and tunable control over both computational efficiency and subject fidelity. The impact of its mask threshold $\epsilon$ serves as a clear

demonstration of this capability. In Figure 10, at the baseline setting (w/o KSA, equivalent to $\epsilon$=0), the model ensures maximum subject fidelity but at a significant computational cost of 16.59s in latency and 368MB of VRAM consumption.

As the threshold is increased to 0.4, KSA strategically prunes the attention map to yield substantial efficiency gains, reducing latency and VRAM to just 15.26s and 242MB, respectively. Notably, even at this more aggressive setting, the generated image remains highly faithful to the reference. The differences are confined to subtle variations in fine details, such as the rendering of the chair's legs and the motorcycle's windshield, showcasing a graceful trade-off rather than an abrupt drop in quality. This highlights the robustness of KSA to its threshold; it is not a sensitive hyperparameter but an intuitive control that allows users to freely balance computational savings with the precise level of subject fidelity their application requires.

### 4.3.3 EFFECT OF EARLY-TIMESTEP SAMPLING

Figure 11 visually demonstrates the effectiveness of our early-timestep sampling strategy. Prioritizing early timesteps (a positive $\mu$) yields markedly superior outcomes for visual condition fine-tuning compared to the standard ($\mu = 0$) or late-biased (a negative $\mu$) approaches. Our proposed early-timestep sampling not only accelerates the convergence of the fine-tuning process but also leads to a final model with enhanced control fidelity.

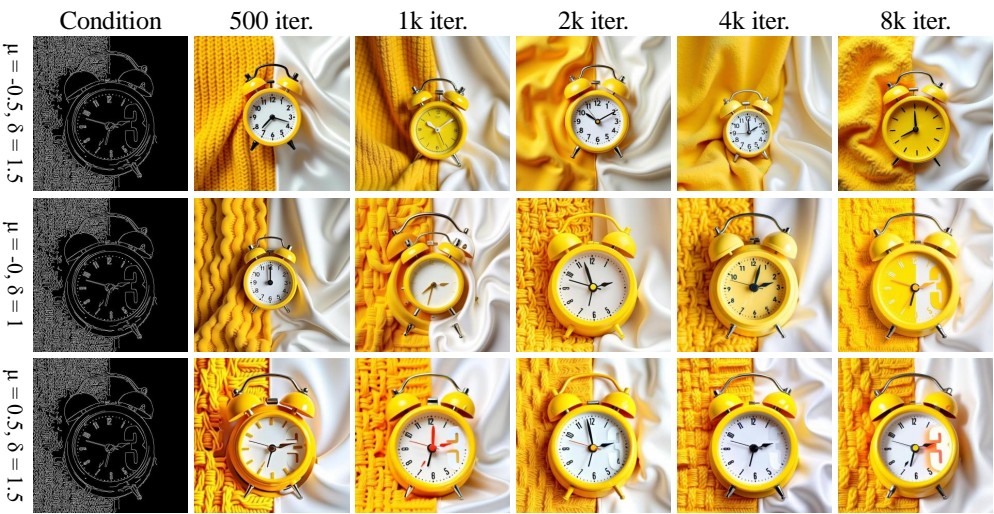

Figure 11: Comparison of visual conditional generation results across different $\mu$ and $\delta$.

## 5 CONCLUSION

In this paper, we addressed the computational inefficiency of multi-condition Diffusion Transformers by proposing Patch-wise and Keyword-Aware Attention (PKA), a novel mechanism that decomposes full attention into two efficient modules: Position-Aligned Attention (PAA) for spatial conditions and Keyword-Scoped Attention (KSA) for subject-driven ones. Our extensive experiments validate this approach, demonstrating a significant up to $10.0\times$ inference speedup and a $5.12\times$ reduction in VRAM consumption for the attention module, all while maintaining or even enhancing generative quality and controllability compared to state-of-the-art methods. Looking ahead, the significant efficiency gains of multi-condition control of PKA make it a promising foundation for tackling more complex generative tasks. A particularly exciting future direction is extending our framework to video generation, where PKA's principles could be applied to enforce temporal consistency across frames at a manageable computational cost. Ultimately, PKA offers a scalable and practical solution that paves the way for the next generation of complex and resource-friendly AI applications.

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

# A   APPENDICES

## A.1   USAGE OF LARGE LANGUAGE MODELS

In adherence to the ICLR 2026 disclosure policy, we report the use of a Large Language Model (LLM) during the preparation of this manuscript. The role of the LLM was strictly limited to that of a writing aid for polishing and proofreading. The authors first drafted the entire content, including the methodology, results, and conclusions. Subsequently, specific sections of the pre-written text were processed by the LLM to identify and suggest corrections for grammatical errors, spelling mistakes, and awkward phrasing.

All suggestions generated by the LLM were critically reviewed by the authors, who retained full control and made the final decision on whether to accept, modify, or reject the proposed changes. The LLM did not contribute in any way to the core scientific ideas, experimental design, data analysis, or the formulation of conclusions presented in this work. The intellectual contribution, conceptual framework, and all scientific claims are entirely the work of the human authors, who bear full responsibility for the content of this paper.

## A.2   VISUAL CONDITION PERTURBATION

To investigate the temporal influence of the visual condition, we conduct a perturbation analysis on Ominicontrol, where the condition is removed at different timesteps during the denoising process. We compare two sequences: a 'high-to-low' order, where perturbation starts from the early, high-noise timesteps (high $t$), and a 'low-to-high' order, which does the reverse.

Figure 12 offers a stark qualitative comparison of these two scenarios. In the 'high-to-low' sequence, the generated images rapidly lose coherence with the visual condition. The core structure and key subject features begin to diverge significantly after only a few perturbed steps. In stark contrast, when applying perturbations in the 'low-to-high' order, the outputs remain remarkably faithful to the condition for a much longer duration, with major deviations only appearing near the end of the process. This visual evidence strongly supports our conclusion: the visual condition exerts its most critical influence and establishes the foundational structure of the image during the initial phase of the generation process.

## A.3   CONVERGENCE OF EARLY-TIMESTEP SAMPLING

Prevailing flow matching models typically adopt a standard logit-normal timestep sampling strategy, where the timestep $t$ is drawn from a Logit-$\mathcal{N}(0, 1)$ distribution to ensure the model trains across the full generation trajectory. Building on our insight that visual conditions are most influential early in this process, we propose an early-timestep sampling strategy. We modify the sampling distribution to a shifted logit-normal, $t \sim \text{Logit-}\mathcal{N}(\mu, \delta)$, where setting $\mu > 0$ intentionally prioritizes these critical early phases.

Figure 13 demonstrates the clear advantage of this approach by plotting the SSIM score during training for different sampling strategies. Our proposed strategy with $\mu = 0.5$ and $\delta = 1.5$ (the orange line) achieves a significantly faster convergence rate and reaches a higher final SSIM score compared to both the standard strategy where $\mu = 0$ (blue line) and a strategy biased towards later timesteps where $\mu = -0.5$ (green line). This confirms that our targeted sampling strategy not only accelerates the training process but also leads to a better-converged model with superior final performance.

## A.4   SCALABILITY WITH INCREASING CONDITIONS

Figure 14 showcases successful generations using two simultaneous conditions, such as combining a subject with a sketch. The complexity is increased in Figure 15, which presents high-quality results from three conditions, and is further demonstrated in Figure 16, which shows robust generation under four simultaneous conditions.

Across these examples, our method consistently and harmoniously synthesizes the multiple constraints, maintaining high visual quality and strong fidelity to each input condition. This highlights

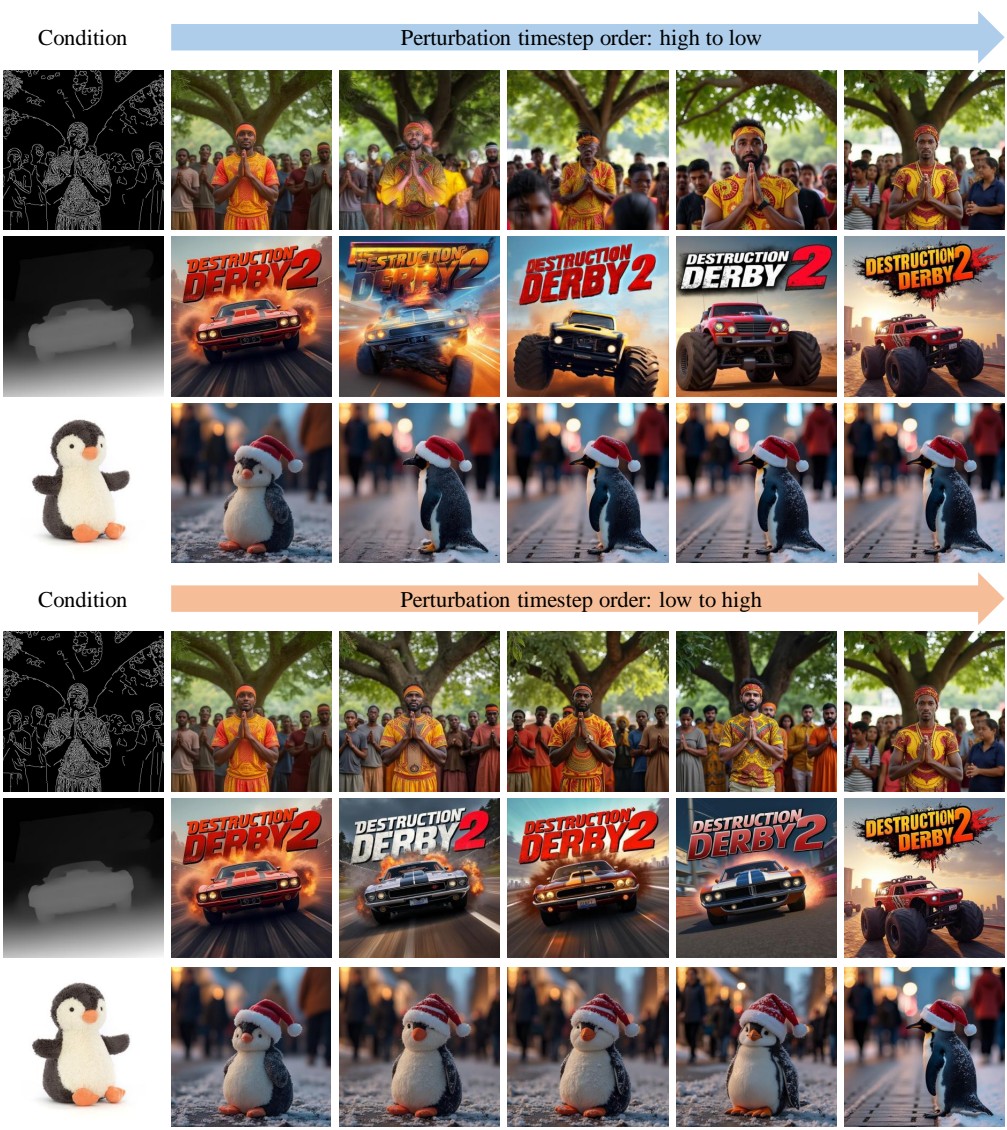

Figure 12: Qualitative results of visual condition perturbation. Left to right: visual condition, 0 (no perturbation), 7, 14, 21 perturbation steps, and 28 steps (no visual condition). Zoom in for better visualization.

the scalability and effectiveness of our approach in handling complex, multi-conditional generation scenarios.

## A.5 MORE QUALITATIVE COMPARISON WITH BASELINES

Figure 17 displays a qualitative comparison of our method against the OminiControl2 and UniCombine baselines across a variety of challenging multi-conditional tasks. The images generated by OminiControl2 often suffer from low visual quality and contain noticeable artifacts. While UniCombine's results are more structurally coherent, they frequently exhibit a muted or desaturated color palette and demonstrate weaker adherence to the provided visual conditions. In contrast, our proposed method consistently produces high-quality images with rich, vibrant colors. More importantly, our approach shows superior fidelity, accurately rendering both the specified subject and the detailed spatial constraints from the conditions.

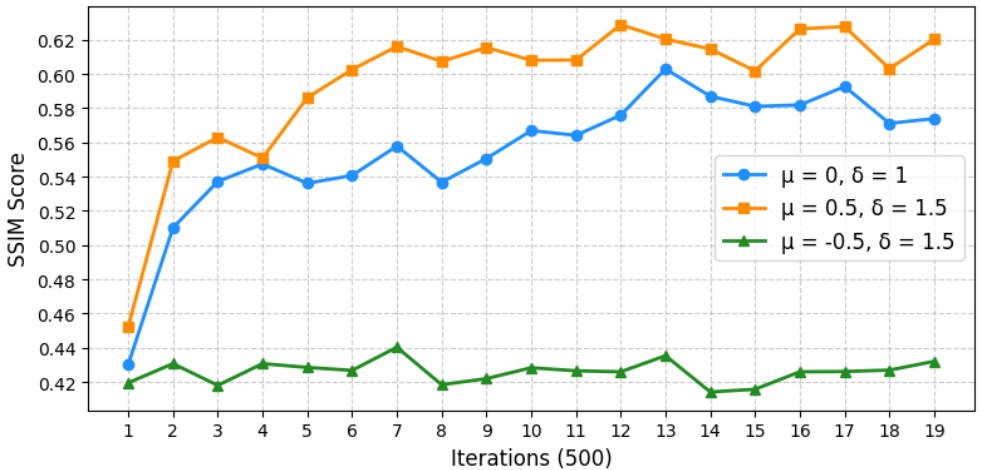

Figure 13: SSIM across the training iteration. our early-timestep sampling ($\mu = 0.5, \delta = 1.5$) achieves better convergence.

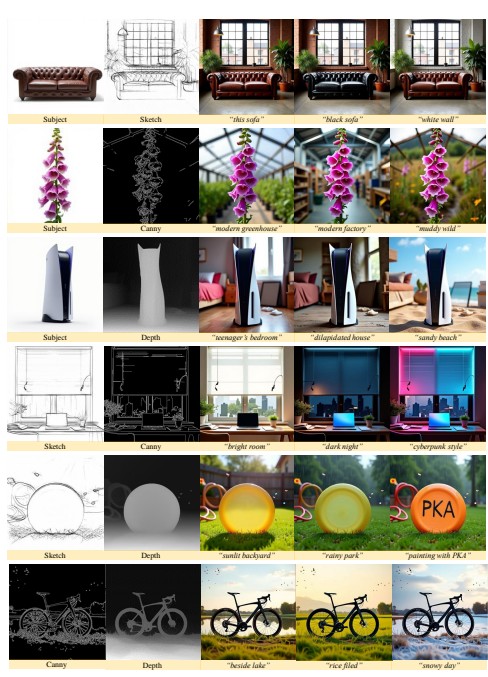

Figure 14: The images generated by 2 conditions. Zoom in for better visualization.

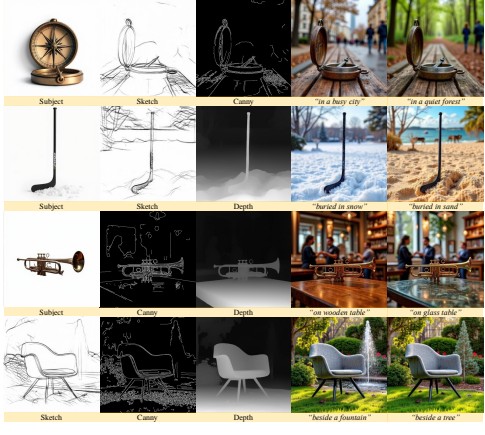

Figure 15: The images generated by 3 conditions. Zoom in for better visualization.

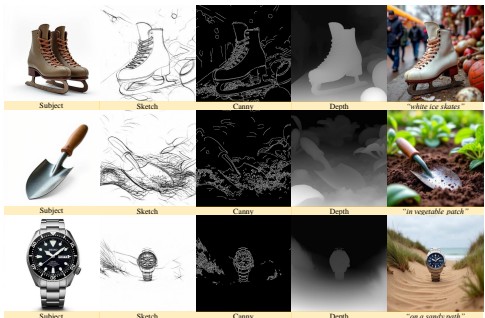

Figure 16: The images generated by 4 conditions. Zoom in for better visualization.

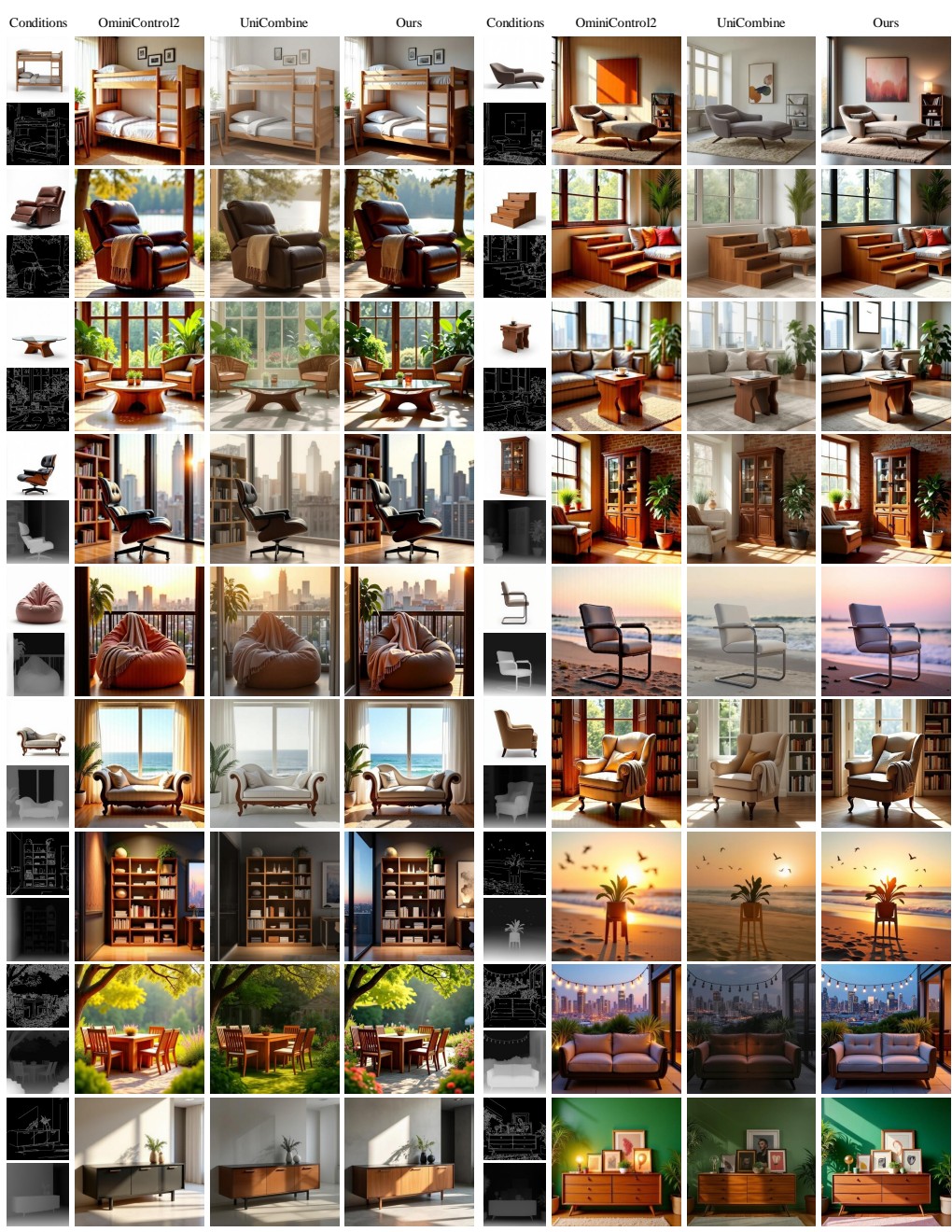

Figure 17: Visual comparison on Subject-Canny-to-Image, Subject-Depth-to-Image, and Canny-Depth-to-Image. Zoom in for better visualization.

