# OpenReview forum: "Patch-Wise and Keyword-Aware: Efficient Multi-Condition Control of Diffusion Transformers via Position-Aligned and Keyword-Scoped Attention"
_ICLR.cc/2026/Conference — ICLR 2026 Conference Withdrawn Submission_

### Official Review · Reviewer_bPZi · 2025-10-19

**Soundness:** 2
**Presentation:** 3
**Contribution:** 2
**Rating:** 2
**Confidence:** 3

**Summary:**

The paper addresses multi-condition Diffusion Transformers’ (DiTs) inefficiency from “concatenate-and-attend” (quadratic cost) by proposing PKA (Patch-wise and Keyword-Aware Attention). It finds attention redundancy in spatial (diagonal focus) and subject-driven (keyword-localized) conditions, then designs two modules: PAA and KSA. It enables efficient fine-grained multi-condition control, paving the way for resource-friendly AI generation (e.g., video) where complexity was prohibitive.

**Strengths:**

1. It pioneers systematic analysis of attention redundancy in multi-condition DiTs, categorizing it into spatial-aligned and subject-driven types.

2. Comprehensive experiments validate efficacy—up to 10× speedup and 5.12× VRAM reduction vs. baselines (OminiControl2, UniCombine) across 3 tasks.

**Weaknesses:**

1. Experiments rely on a Subject200K subset with simple single-subject tasks (Subject-Canny/Depth-to-Image) but lack testing on complex scenarios (multi-subject, heavy occlusion, low-resource domains like medical/industrial tables). This limits generalization validation.

2. It only compares with OminiControl2/UniCombine, omitting recent efficient DiT methods that also optimize attention/memory. Actually this paper is not the first and not the only paper focusing on DiT-based multi-conditional generation. UniCombine[1], PixelPonder[2] and also [3] can be referred to. Further comparisons with these new methods and also the UniControl[4], UniControlNet[5], CockTail[6], Ctrl-X[7] are encouraged. The most important thing is to differ from the previous solutions and show your advantages over them.

[1] UniCombine: Unified Multi-Conditional Combination with Diffusion Transformer (ICCV 2025 oral)

[2] PixelPonder: Dynamic Patch Adaptation for Enhanced Multi-Conditional Text-to-Image Generation

[3] Context-Aware Autoregressive Models for Multi-Conditional Image Generation

[4] UniControl: A Unified Diffusion Model for Controllable Visual Generation In the Wild (NeurIPS 2023)

[5] Uni-ControlNet: All-in-One Control to Text-to-Image Diffusion Models (NeurIPS 2023)

[6] Cocktail: Mixing Multi-Modality Controls for Text-Conditional Image Generation

[7] Ctrl-X: Controlling Structure and Appearance for Text-To-Image Generation Without Guidance (NeurIPS 2024)

**Questions:**

refer to the weaknesses.

---

### Official Review · Reviewer_hEJs · 2025-10-21

**Soundness:** 2
**Presentation:** 3
**Contribution:** 2
**Rating:** 2
**Confidence:** 5

**Summary:**

This paper addresses the critical computational inefficiency of multi-condition control in Diffusion Transformers (DiTs), a limitation imposed by the dominant "concatenate-and-attend" strategy—where text, noisy image, and visual condition tokens are jointly processed via full attention, leading to quadratic scaling of compute and memory costs with the number of conditions. The authors first conduct an empirical analysis of attention patterns in multi-condition DiTs, identifying two key forms of redundancy: (1) spatial-aligned conditions (e.g., depth maps, Canny edges) exhibit localized attention concentrated on spatially aligned patches; (2) subject-driven conditions (e.g., reference subjects, text keywords) activate only a small subset of image tokens relevant to the target subject.
To eliminate this redundancy, the paper proposes Patch-wise and Keyword-Aware Attention (PKA), a lightweight framework with three core components. Experiments are conducted on a subset of the Subject200K dataset, fine-tuning the FLUX.1 model with LoRA. Results show PKA achieves up to 10× inference speedup and 5.12× reduction in attention module VRAM consumption compared to baselines (OminiControl2, UniCombine), while maintaining or improving generative quality (e.g., 52.99 FID vs. 61.03 for UniCombine on Subject-Canny task) and controllability (e.g., 0.553 SSIM vs. 0.493 for UniCombine on Subject-Canny task).

**Strengths:**

- Unlike generic efficiency methods (e.g., token pruning in PixelPonder, layer caching in FastCache), PKA leverages condition-specific structural priors (spatial locality for layout conditions, keyword relevance for subject conditions) to prune attention. This design is more principled than one-size-fits-all strategies, as it directly addresses the root cause of inefficiency in multi-condition DiTs.
- PKA integrates three complementary efficiency mechanisms: PKA’s attention pruning reduces per-step compute, the condition cache eliminates redundant Key/Value projections across steps, and early-timestep sampling accelerates training. This end-to-end optimization outperforms baselines (e.g., 3.90×–10× speedup over UniCombine) while avoiding quality trade-offs—a gap in existing methods like OminiControl2 (which suffers from visual artifacts) or UniCombine (which produces desaturated colors).
- The authors first validate attention redundancy via perturbation studies (Figure 5, Figure 12), showing visual conditions exert maximal influence in early denoising stages. This empirical foundation ensures PKA’s design is data-driven, rather than heuristic—unlike some prior efficiency methods (e.g., ad-hoc token pruning in Zou et al., 2025) that lack such validation.
- Experiments cover three multi-condition tasks (Subject-Canny-to-Image, Subject-Depth-to-Image, Canny-Depth-to-Image) and use a comprehensive set of metrics: efficiency (latency, VRAM), generative quality (FID, SSIM), controllability (F1 for edges, MSE for depth), subject consistency (CLIP-I, DINOv2), and text alignment (CLIP-T). Ablation studies further isolate the impact of PAA (vs. full attention/Sliding Window Attention) and KSA (mask threshold tuning), strengthening the credibility of PKA’s effectiveness.

**Weaknesses:**

- The paper only compares PKA against OminiControl2 (Tan et al., 2025) and UniCombine (Wang et al., 2025), while omitting key competitors in multi-condition DiT control. Most notably, it excludes PixelPonder (Pan et al., 2025) (dynamic patch adaptation for multi-condition generation) and UniControl (a widely cited multi-modal control framework for DiTs)—both of which address efficiency via condition-aware token adaptation, directly relevant to PKA’s goals. Without these comparisons, PKA’s competitiveness in the broader multi-condition control landscape remains unproven.
- KSA relies on a fixed mask threshold $\epsilon=0.2$ (Section 3.2.2) for all tasks and datasets, but the paper provides no analysis of how $\epsilon$ impacts performance across different condition types (e.g., fine-grained subjects vs. coarse spatial layouts) or datasets. It also does not validate whether $\epsilon$ requires retuning for new conditions—leaving uncertainty about KSA’s ease of use in practical scenarios.
- PKA maintains full attention between noisy image tokens and text tokens (Section 3.2), but the paper does not analyze how text-keyword relevance (used in KSA) interacts with other conditions (e.g., whether text conflicts with spatial layouts). Existing methods like GLIGEN (Li et al., 2023) explicitly model text-spatial alignment, but PKA’s handling of such interactions is unaddressed—limiting understanding of its robustness to conflicting multi-condition inputs.
- The multi-condition tasks only include three modalities (subject references, Canny edges, depth maps), excluding other common multi-condition scenarios: (1) semantic masks (e.g., segment-level control), (2) human poses (e.g., pose-guided subject generation), and (3) style references (e.g., artistic style transfer). PKA’s effectiveness for these underrepresented conditions—where attention patterns may differ (e.g., style spans global image regions)—remains untested.

**Questions:**

- Why were critical SOTA multi-condition control methods like PixelPonder (Pan et al., 2025) (dynamic patch adaptation) and UniControl (multi-modal DiT control) excluded from baseline comparisons? How does PKA’s efficiency (speed, VRAM) and quality (controllability, fidelity) compare to these methods, especially for tasks with 4+ conditions?
- Does the KSA mask threshold $\epsilon$ need retuning for different condition types (e.g., fine-grained subjects vs. coarse depth maps) or - datasets (e.g., Subject200K vs. COCO)? What performance degradation occurs if a fixed \(\epsilon\) is used across diverse scenarios, and is there an adaptive way to set $\epsilon$ automatically?
- The paper demonstrates PKA’s effectiveness for up to 4 conditions (Figure 16), but how does it perform with 5+ conditions (e.g., subject + depth + Canny + semantic mask + style reference)? Does the efficiency gain (speedup/VRAM reduction) diminish with more conditions, and if so, what is the bottleneck?

---

### Official Review · Reviewer_x4EF · 2025-10-31

**Soundness:** 3
**Presentation:** 3
**Contribution:** 2
**Rating:** 4
**Confidence:** 4

**Summary:**

The manuscript is an extension of an existing multi-condition DiT method. The proposed method focuses on reducing dense self-attention in the previous method based on “modality concatenation” (OminiControl). Based on the observation that the attention map is sparse between the condition and denoising tokens, the authors 1) restrict attention from denoising tokens to condition tokens to a local region (PAA), and 2) restrict the denoising tokens that only align with a “keyword” to receive information from text through attention. The author further pinpoints that early timesteps are more effective in achieving conditioning, thus using a skewed timestep scheduler to further improve efficiency. The writing is clear and easy to follow. However, the choice of the dense self-attention model over cross-attention models is unclear.

**Strengths:**

The problem of increasing the efficiency of multi-condition diffusion models is important regarding deploying such models. The proposed method of attention pruning is straightforward. It is expected to see it works well. The result of effectiveness is convincing and supported by ablation analysis.

**Weaknesses:**

**Overly emphasized “concatenation” models**

Not all the multi-condition models adopt “concatenate”, thus the motivation that “concatenate-and-attend is computationally prohibitive” is weak. The complexity of O(c^2n^2) is only true for models such as OminiControl. However, there are a considerable number of models that do not use this design for multi-modality conditioning but use more efficient cross attention, such as IP-Adapter and UNIC-Adapter. They should be introduced and discussed in the main text (concatenation v.s. cross-attention, pros and cons, etc.).

**Weak control**

The proposed method is only compared to two other models (OminiControl2 and UniCombine). Well-known models such as IP-Adapter are supposed to be compared, especially given that they have different attention designs with cross-attention. It is possible IP-Adapter is more efficient than “pruned” concatenation models.

**Inconsistent language**

The proposed method is named “patch-wise and keyword-aware attention” (PKA), but the titles of 3.2.1 and 3.2.2 are “position-aligned attention” (PAA) and “keyword-scoped attention” (KSA). They seem to be the exact components of PKA, but the names are changed? If they mean different things, what is the difference between “patch-wise” and “position-aligned”.

**Questions:**

Models like PixArt popularize the use of cross-attention to achieve conditioning. Authors may explain what are the advantages of using dense self-attention design. Especially given that PixArt is known to be efficient, is pruning the dense self-attention a promising direction? Since the proposed PAA and KSA are even more localized than cross-attention, they are applicable to PixArt. Or is it fair to say a concatenation model with PAA and KSA is similar to using cross-attention?

---

### Official Review · Reviewer_2wnk · 2025-11-01

**Soundness:** 2
**Presentation:** 2
**Contribution:** 2
**Rating:** 4
**Confidence:** 3

**Summary:**

This paper proposes PKA, a two-component attention mechanism --- Position-Aligned Attention (PAA) and Keyword-Scoped Attention (KSA) to reduce computational overhead in multi-condition DiTs. It also introduces an early-timestep sampling strategy for fine-tuning. The method is evaluated on FLUX.1 with LoRA, reporting up to 10× speedup and 5.12× VRAM reduction. The results show that the proposed method achieve better qualitative and quantitative performance as compared to the existing baseline methods.

**Strengths:**

1. The paper empirically shows an observation that attention in multi-condition DiTs is highly sparse and structured  as shown in figure 2, 3.  The authors demonstrate that attention is highly structured and sparse, with spatial conditions exhibiting diagonal-localized interactions and subject-driven conditions activating only keyword-relevant regions.
2. The proposed PAA for spatial conditions and KSA for subject references enhances interpretability and also facilitates extensions to other condition types.
3. The method delivers impressive computational benefits, up to 10× faster inference and 5.12× lower VRAM usage in the attention module without compromising generation quality.
4. The authors compare against recent baselines (OminiControl2, UniCombine), and conducted ablation studies on various components.

**Weaknesses:**

1. Though, the results shown in the paper are impressive, it seems to me that the proposed method is incrementally driven from existing methods. PAA is essentially a restatement of spatial locality assumptions already exploited in OminiControl and refined in OminiControl2, which uses “minimal universal control” and “efficient token interaction” for spatial conditions. OminiControl2 explicitly reduces conditional processing overhead by >90% through dynamic token pruning and alignment, functionally equivalent to PAA’s one-to-one patch matching.

2. The paper assumes that 1 or 2 tokens can be reliably identified as “keywords” (e.g., “shirt”) to drive KSA. However no algorithm is provided for keyword extraction (manual? POS tagging? attention scores?). This mirrors earlier methods like Attend-and-Excite (SIGGRAPH), which use attention-based semantic guidance but PKA offers no improvement or analysis over these. Without a robust, automated keyword localization strategy, KSA’s applicability is limited to curated prompts, undermining its practical utility.

3. All experiments use FLUX.1 fine-tuned with LoRA which narrows setup that does not demonstrate generalizability. Given that OminiControl2 and UniCombine are architecture-agnostic frameworks, PKA’s restriction to one model raises concerns about overfitting to implementation quirks rather than proposing a general solution.

**Questions:**

1. How does PAA differ technically from OminiControl2’s spatial token alignment and pruning mechanism?
2. Can KSA’s keyword selection be automated without human annotation or heuristic thresholds?
3. Why was no comparison made against PixelPonder, which also uses dynamic region selection for multi-condition control?
4. Have you tested PKA on non-FLUX DiTs (e.g., SD3/3.5) to validate architectural generality?
5. Does the early-timestep sampling strategy degrade performance on tasks requiring late-stage refinement?

---

### Note · Authors · 2025-11-12

I have read and agree with the venue's withdrawal policy on behalf of myself and my co-authors.